# Improved digital chest tomosynthesis image quality by use of a projection-based dual-energy virtual monochromatic convolutional neural network with super resolution

**Tsutomu Gomi** [ORCID]*, **Hidetake Hara, Yusuke Watanabe, Shinya Mizukami**

School of Allied Health Sciences, Kitasato University, Sagamihara, Kanagawa, Japan

* gomi@kitasato-u.ac.jp

**Data Availability Statement:** All relevant data are within the manuscript and its Supporting Information files.

## Abstract

We developed a novel dual-energy (DE) virtual monochromatic (VM) very-deep super-resolution (VDSR) method with an unsharp masking reconstruction algorithm (DE–VM–VDSR) that uses projection data to improve the nodule contrast and reduce ripple artifacts during chest digital tomosynthesis (DT). For estimating the residual errors from high-resolution and multi-scale VM images from the projection space, the DE–VM–VDSR algorithm employs a training network (mini-batch stochastic gradient-descent algorithm with momentum) and a hybrid super-resolution (SR) image [simultaneous algebraic reconstruction technique (SART) total-variation (TV) first-iterative shrinkage–thresholding algorithm (FISTA); SART–TV–FISTA] that involves subjective reconstruction with bilateral filtering (BF) [DE–VM–VDSR with BF]. DE-DT imaging was accomplished by pulsed X-ray exposures rapidly switched between low (60 kV, 37 projection) and high (120 kV, 37 projection) tube-potential kVp by employing a 40° swing angle. This was followed by comparison of images obtained employing the conventional polychromatic filtered backprojection (FBP), SART, SART–TV–FISTA, and DE–VM–SART–TV–FISTA algorithms. The improvements in contrast, ripple artifacts, and resolution were compared using the signal-difference-to-noise ratio (SDNR), Gumbel distribution of the largest variations, radial modulation transfer function (radial MTF) for a chest phantom with simulated ground-glass opacity (GGO) nodules, and noise power spectrum (NPS) for uniform water phantom. The novel DE–VM–VDSR with BF improved the overall performance in terms of SDNR (DE–VM–VDSR with BF: 0.1603, without BF: 0.1517; FBP: 0.0521; SART: 0.0645; SART–TV–FISTA: 0.0984; and DE–VM–SART–TV–FISTA: 0.1004), obtained a Gumbel distribution that yielded good images showing the type of simulated GGO nodules used in the chest phantom, and reduced the ripple artifacts. The NPS of DE–VM–VDSR with BF showed the lowest noise characteristics in the high-frequency region (~0.8 cycles/mm). The DE–VM–VDSR without BF yielded an improved resolution relative to that of the conventional reconstruction algorithms for radial MTF analysis (0.2–0.3 cycles/mm). Finally, based on the overall image quality, DE–VM–VDSR with BF improved the contrast and reduced the high-frequency ripple artifacts and noise.

**Funding:** This study was supported by a grant from Kitasato University School of Allied Health Sciences (Grant-in-Aid for Research Project, No. 2020-1006).

**Competing interests:** The authors have declared that no competing interests exist.

## Introduction

Lung cancer is the leading cause of cancer deaths, and its incidence has considerably increased globally [1–6]. Although conventional helical computed tomography (CT) is accepted as the gold standard because of its high level of sensitivity for detecting lung cancer, early-stage lung cancer can be detected via low-dose helical CT, helping to reduce morbidity. However, CT has certain drawbacks compared with chest radiography, including elevated cost and radiation dose. The advantages of chest radiography over CT are that examinations are much shorter and easily obtained; however, the former suffers from lower specificity and sensitivity. In chest X-ray radiography, a three-dimensional structure of the chest is projected as a two-dimensional image. Thus, overlapping anatomical structures can occasionally obscure the features necessary for diagnosis based on chest X-ray images.

Pulmonary-nodule detection and characterization based on chest imaging are difficult tasks. High-resolution CT (HR-CT) reveals the comprehensive organization of local and diffuse substantive aberrations and enables detection of their anatomical distribution that has considerably improved the ability to evaluate nodules [4, 7, 8]. Digital tomosynthesis (DT) provides many of the advantages associated with digital imaging, including low-dose short-duration examinations, partial volume effect (PVE)-free longitudinal-direction images, and low-cost availability [9–12]. DT enhances the ability to detect abnormalities by removing the overlapping anatomical structures and refining the clarity and sharpness of the in-plane structures in images [13].

The dual-energy subtraction (DES) technology can be used in DT chest imaging to enhance detection relative to that of conventional polychromatic imaging [14, 15]. Although DES can produce various images of soft tissue and bone, it increases noise [12, 14]. However, DES-DT generates a great "rippled" artifact on any visible pulmonary nodules because of the intrinsic misalignment of low- and high-kVp images [14, 16]. In chest DT (CDT), reconstruction from a limited arc angle has been reported to cause ripples, increasing the difficulty of precise diagnosis [14, 16]. Because of this rippled pattern, we anticipate that these pulmonary nodules may not be noticed and that detection of the pulmonary nodules can be improved if this rippled pattern can be avoided.

Earlier reports indicated the use of DE-CT material decomposition and virtual monochromatic (VM) imaging (projection-space approach and image-space approach) to enhance nodule detection [17–20]. Particularly, enhancements in nodule detection were obtained via reconstruction of projection data decomposed using the materials in projection space [20, 21]. Low- and high-energy types of weighted blending methods [22, 23] are involved in the two-material decomposition-based DE technique. Depending on the selected blending ratio, the noise or contrast may be compromised in this technique. Therefore, a blended image with reduced noise but lower contrast and spatial-resolution information is obtained by changing the blending ratio. Nevertheless, these inherent problems potentially can be resolved at a high level by combining the DE technique with three-material decomposition.

Although filtered backprojection (FBP) [13] has excellent high-frequency detection sensitivity, the enhanced noise and ripple artifacts are concerning. In contrast, a simultaneous algebraic reconstruction technique (SART) [24] is expected to be effective at suppressing noise in iterative reconstruction (IR) and has been useful in various fields [25–28]. The usefulness of combining the first-iterative shrinkage–thresholding algorithm (FISTA) and IR–total variation (TV) was recently reported [29, 30]. Integrated and improved images may be generated through the combination of FISTA, IR-TV, and deep learning. Therefore, we adopted a combined reconfiguration, i.e., SART–TV–FISTA, for this study.

Deep learning methodologies have been successfully instituted in pattern recognition and processing of images, including denoising [31–33] and generation of super-resolution (SR)

[34–38] images. For example, a convolutional neural network (CNN) has been applied to enhance the detection of pulmonary nodules in medical imaging by removing residual errors [39–43]. A CNN-based modification (very-deep SR [VDSR]) was presented by Kim et al. [44]. The main feature of VDSR includes the reconstruction progress associated with the learning algorithms. The very-deep architecture combined with regularization procedures increase the contrast and resolution of the image. The reference and residual images used in the training workflow for VDSR are essential for improving the contrast and resolution of the nodule.

In DT, nodule detection can be enhanced by improving the clarity and sharpness of the DE–VM projection images. Therefore, we recommend applying CNN-based deep-learning processing to the DE–VM projection images for improving their resolution (so-called SR). Enhancing the image resolution by SR is useful for more accurate detection of nodules. SR is a method for generating high-resolution (HR) and low-resolution (LR) images. Single-image SR is difficult because high-frequency image content cannot be obtained from an LR image. The HR image quality is limited in the absence of high-frequency information. VDSR is a CNN architecture developed to perform single-image SR processing [44]. The VDSR network learns the mapping between LR and HR images. These images differ mainly in terms of the high-frequency details even though they have similar image content, making this mapping possible. Digital enlargement (zoom in) of a poor image results in image blurring and poor image quality because the conventional interpolation or enlargement algorithm is insufficient. However, a high-quality HR image can be generated from an LR image via VDSR.

DE–VM spectral imaging can potentially improve nodule detection [17–20]. We postulate that DE–VM will enhance the nodule contrast and improve nodule detection. We anticipate an increase in nodule contrast and resolution after reconstruction when VDSR processing is used to conduct residual learning at the projection-data level. To support this proposition, VDSR, which constitutes a built-in deep feedforward CNN, is employed to produce an SR image from the residual image. However, because VDSR uses a residual learning method, it is also feasible to train the VDSR network architecture for increasing the nodule contrast and resolution. Furthermore, the nodule contrast may be augmented through reconstruction in which the decomposed projection data from every material (e.g., nodule, soft tissue, vessel, and bone) processing are considered. In case of DE, an increase in ripple artifacts with noise (especially high frequency) because of the influence of VM processing is concerning. In addition, detection by VDSR is expected to be difficult for ground-glass opacity (GGO) nodules associated with low X-ray absorption. As a countermeasure, we proposed correcting VDSR with unsharp masking (UM [45]) for the projection space (preprocessing) and the reconstructed image (postprocessing) with bilateral filtering (BF) [26]. We believe that VDSR with UM and BF is useful for contrast enhancement and reducing ripple artifacts because it can reduce high-frequency noise while preserving the edge information of soft tissues and nodules.

The aim of this study was to describe a newly developed projection-space hybrid reconstruction procedure by employing DE–VM with VDSR to increase the nodule contrast and reduce ripple artifacts [DE–VM with VDSR (DE–VM–VDSR with BF)] from CDT.

## Materials and methods

### Overview of DE–VM–VDSR with BF

The novel DE–VM–VDSR with BF algorithm was implemented in case of projection-space data to increase the nodule contrast while reducing ripple artifacts during CDT. This method is based on a training network (mini-batch gradient-descent algorithm with momentum [SGDM]) with UM to estimate the residual image from HR and multiscale VM images in a projection space involving hybrid and subjectively reconstructed SR images (SART–TV–

FISTA) and BF (Fig 1). The flowchart shows the interrelations of VDSR with UM, SART–TV–FISTA, and BF, which form the core of DE–VM–VDSR with BF (Fig 2).

## Phantom specifications

The Chest Phantom N1 (Kyoto Kagaku Co., Tokyo, Japan) used in this study comprised soft tissue and vessels made of polyurethane resin and artificial material made of epoxy resin and $CaCO_3$. The simulated pulmonary nodules used in this study were the GGO type (−630 Hounsfield units, 5 mm in diameter, urethane foam) when considering a homogenous composition. They were arranged in the right-middle-lobe region. The simulated nodules placed close to the edges of the lungs or combined with the blood vessels were evaluated.

## DE-DT system

The DE-DT system (SonialVision Safire II; Shimadzu Co., Kyoto, Japan) comprised an X-ray tube (anode, made of tungsten, rhenium, and molybdenum; real filter: inherent; aluminum [1.1 mm], additional; aluminum [0.9 mm] and copper [0.1 mm]) with a 0.4-mm focal spot and an amorphous selenium ($1720 \times 1720$ pixels) digital flat-panel detector (detector element, $0.15 \times 0.15$ mm). The source-to-isocenter distance was 924 mm, whereas the source-to-detector distance was 1100 mm (antiscatter grid, focused type; grid ratio, 12:1). The selected kV values (low, 60 kV; high, 120 kV) [14] were optimal for testing the simulated nodule contrast because they inherently offered optimal tissue decomposition.

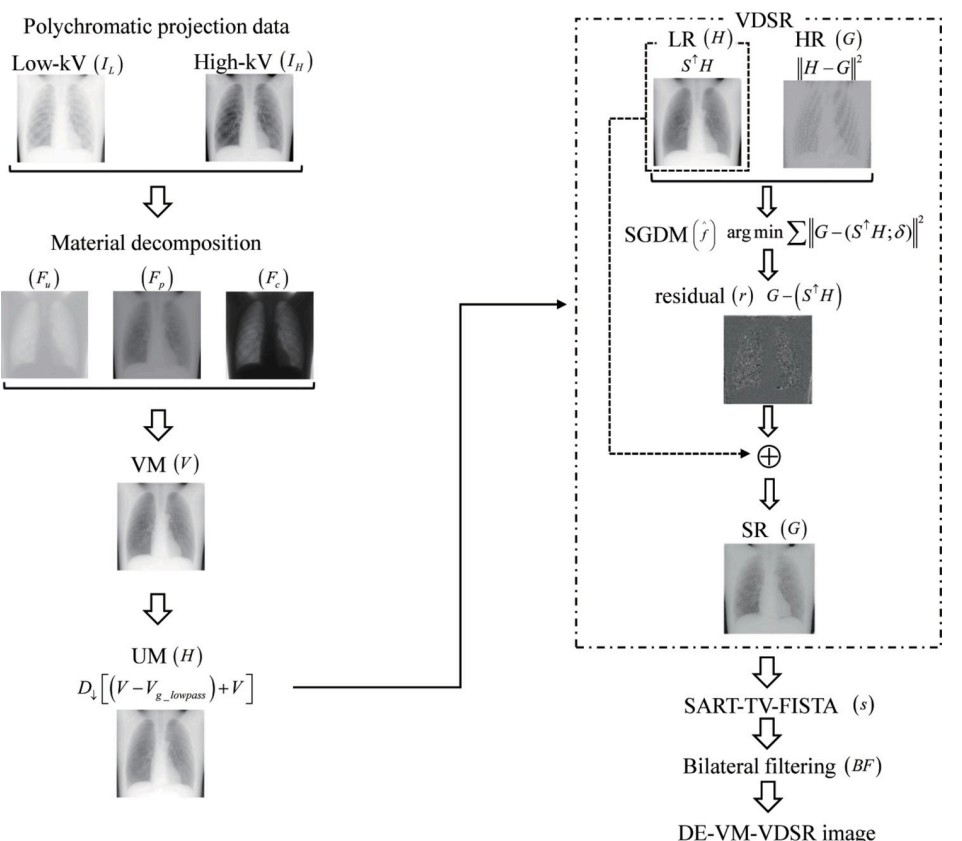

**Fig 1. Flowchart of the Dual-Energy (DE) Virtual Monochromatic (VM) with Very-Deep Super-Resolution (VDSR) reconstruction algorithm (DE–VM–VDSR) and Bilateral Filtering (BF).** DE–VM–VDSR with BF is implemented by combining preprocessing [VM and VDSR with unsharp masking (UM)] and postprocessing (BF).

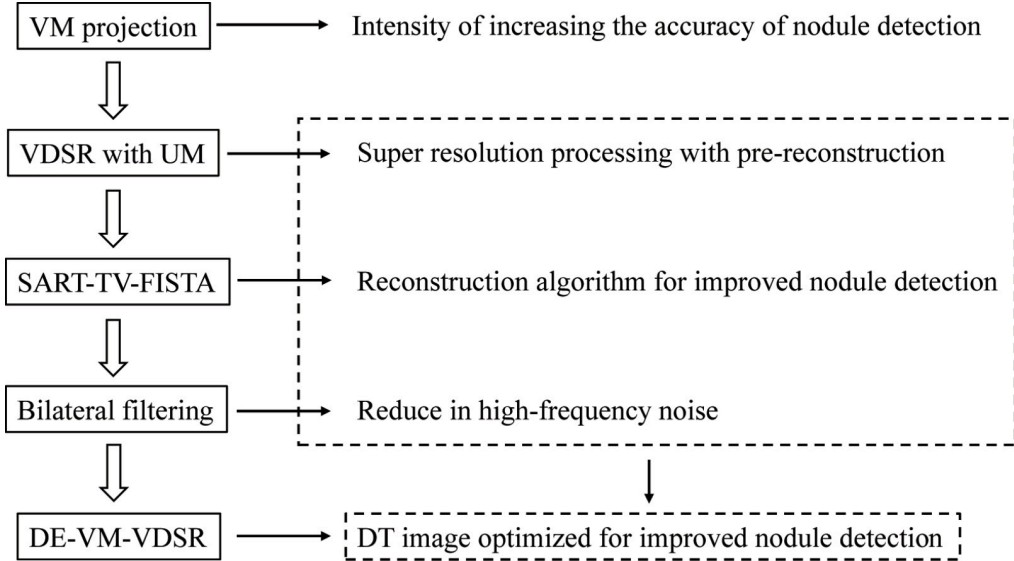

**Fig 2. Overview and interrelation of each algorithm.** The relation between very-deep super-resolution (VDSR) with unsharp masking (UM), the simultaneous algebraic reconstruction technique total-variation first-iterative shrinkage–thresholding algorithm (SART–TV–FISTA), and bilateral filtering (BF), which are the cores of the reconstruction process.

Thirty-seven low- and high-voltage projection images ($1024 \times 1024$ pixels) were obtained at voltages when considering a swing angle of 40˚, a total acquisition time of 6.4 s, and linear system movement during a single tomographic pass. In DE-DT imaging, exposures with pulsed X-rays were employed with fast interchanges between low and high energies even though high voltages are normally used for clinical applications [12, 15]. Using low-voltage X-rays, projection images were obtained at 280 mA with an exposure time of 27 ms, whereas images were obtained using high-voltage X-rays at 416 mA with an exposure time of 2.5 ms (default DE acquisition parameters on the DT device). We used $1024 \times 1024$ pixels with 32 bits (single-precision floating number) per reconstruction image (pixel size, 0.252 mm/pixel; thickness and increments, 1 mm) to recreate the tomograms in the needed longitudinal direction.

## DE–VM–VDSR

**Generation of the DE–VM projection images.**   In this study, we predicted that the generation of VM images via three-material decomposition [46] from DE acquisition could result in the accurate extraction of nodules, especially those of the GGO type. We performed DE–VM processing at the projection-data level to improve contrast, thereby increasing the nodule detection accuracy.

In this study, we used projection-space (pre-reconstruction) decomposition to evaluate the material fractions $F_n$ of the artificial bone ($CaCO_3$ + $C_{15}H_{16}O_2$, $F_c$, local density; 1.3098 g/cm$^3$), soft tissue ($O_4N_2H_2$, $F_p$, local density; 1.0600 g/cm$^3$), and a simulated GGO nodule ($C_3H_8N_2O$, $F_u$, local density; 0.3500 g/cm$^3$) in the phantom.

The linear attenuation coefficient $\mu(E)$ could be calculated for any photon energy $E$ based on the density corresponding to each of the three basic experimental material-containing areas. The theoretical linear attenuation coefficient curve was determined by employing the local and area densities (g/cm$^2$) and mass attenuation coefficient of each material (Fig 3). They are obtained using the XCOM program designed by Berger and Hubbell [47]. Finally, for the projection space, we employed the decomposition approach to produce decomposition images

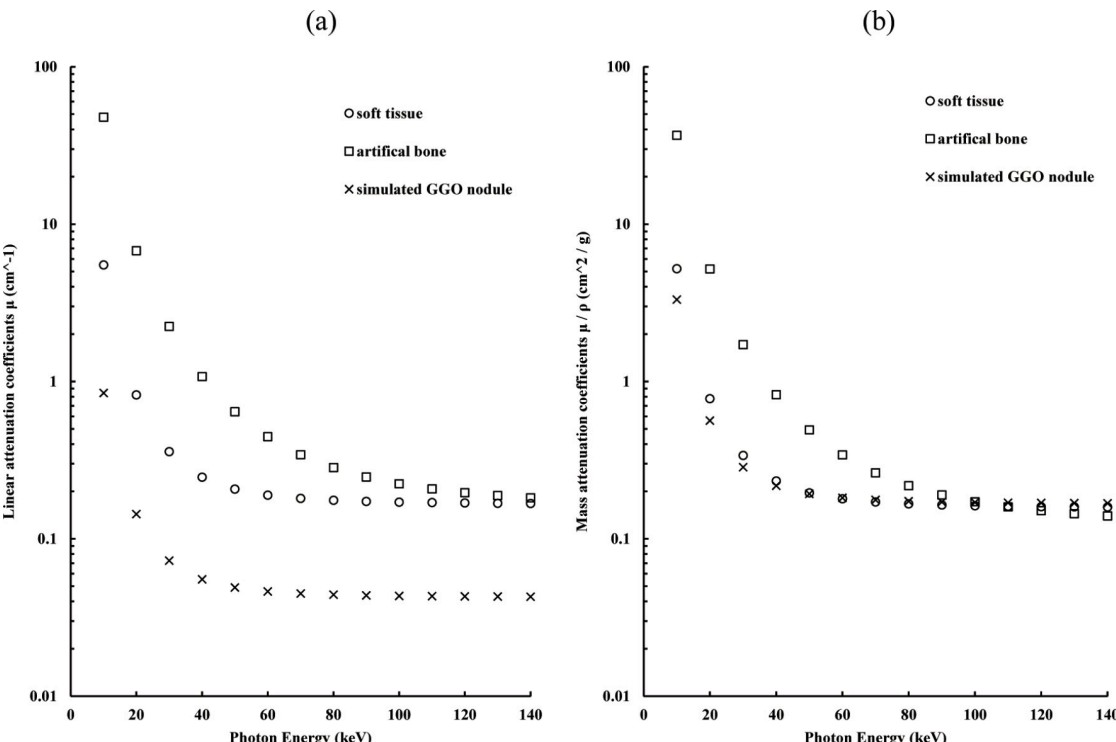

**Fig 3. Linear and mass attenuation coefficients for each energy.** The linear attenuation coefficients of a simulated ground-glass opacity (GGO) nodule, a soft tissue, and an artificial bone with respect to the photon energy.

for materials, including urethane foam, polyurethane, and artificial bone. The material fractions $F_n$ were derived from the inverse of the matrix with the attenuation intensities of three materials at two energy levels. The inverse of this matrix [the "*inv*" function from MATLAB (MathWorks, Natick, MA)] was used. Material fractions were obtained using the above function, which limits the possible fraction to 0 and 1 while imposing a sum of 1. Thus, from the processing pipeline, three-material fractions related to the artificial bone, soft tissue, and simulated GGO nodules can be obtained. DE–VM processing is performed as follows:

$$V = F_u \cdot \left(\frac{\mu}{\rho}\right)_u (E) + F_p \cdot \left(\frac{\mu}{\rho}\right)_p (E) + F_c \cdot \left(\frac{\mu}{\rho}\right)_c (E), \tag{1}$$

$$F_u + F_p + F_c = 1.0,$$

where $V$ is the DE–VM projection image and $(\mu/\rho)_u(E)$, $(\mu/\rho)_p(E)$, and $(\mu/\rho)_c(E)$ are the corresponding mass attenuation coefficients of each material. We have evaluated the optimal DE–VM energy (keV) in the "Optimization parameters" section.

**VDSR.** VDSR exhibits a CNN architecture designed to perform SR processing with residual learning [44]. The VDSR network learns the mapping between LR and HR images. This mapping is possible because LR and HR images have similar image contents and differ primarily in the finer high-frequency components.

**Architecture.** There are 20 weight layers in the network, which are identical, except for the first and last layers. The network comprises 20 conventional layers, each with 64 $3 \times 3$ filters, followed by a rectified linear unit (ReLu) [48]. The desired residual image is reconstructed

in the final layer with a $3 \times 3 \times 64$ filter [44]. Subsequently, this final residual image can be merged with the LR input to obtain the resulting SR output (Fig 4).

**Multiscale.** After the VDSR network learns to assess the residual image, it can recreate HR images by combining the assessed residual image with the upscaled LR image. The HR images were downscaled to produce the corresponding LR images. In this study, scale factors of 2, 3, or 4 were used for training, and the highest contrast was obtained for each scale factor during testing [44]. After downscaling, they were then upscaled to their previous resolution, resulting in a suitably distorted LR image dataset. Data augmentation (rotating to an arbitrarily small angle, shifting an arbitrarily small distance) was applied during training to alter the training data essentially to increase the volume of available training data (by randomly selecting a scale factor as a form of data augmentation). The training model extracted randomized patches from the upscaled and residual images.

**Training dataset.** Thirty-seven reference images [comprising down- and upscaled LR images (bicubic interpolation; scale factors: 2, 3, and 4 [44]) and the corresponding residual images form original projection] related to the revised input image pairs were randomly selected as the training set from the generated VM projection data (total training data set: 74). The original projection images ($1024 \times 1024$) were downsized using different scale factors (2, 3, and 4) [44] to create sample LR images; then, the LR images were resized to their original size via bicubic interpolation. Subsequently, the difference between the original and resized images was determined. The network inputs were LR images upscaled by bicubic interpolation.

**Residual learning.** In this study, we applied UM [45] to the image (original VM image: $V$) before the interpolated LR image ($H$) was processed. The parameter (standard deviation) of UM processing was set to 2.5 to maintain a balance between high-frequency noise and edge preservation. The visibility of the simulated GGO nodule in the projection image was improved by performing UM (Fig 5).

Interpolated LR images are generated as follows:

$$H = D_{\downarrow}[k \cdot (V - V_{g\_lowpass}) + V] \tag{2}$$

$D_{\downarrow}$: Downscaling interpolation operator
$k$: Scaling contrast
$V_{g\_lowpass}$: Gaussian low-pass filtered image
$(V - V_{g\_lowpass}) + V$: UM

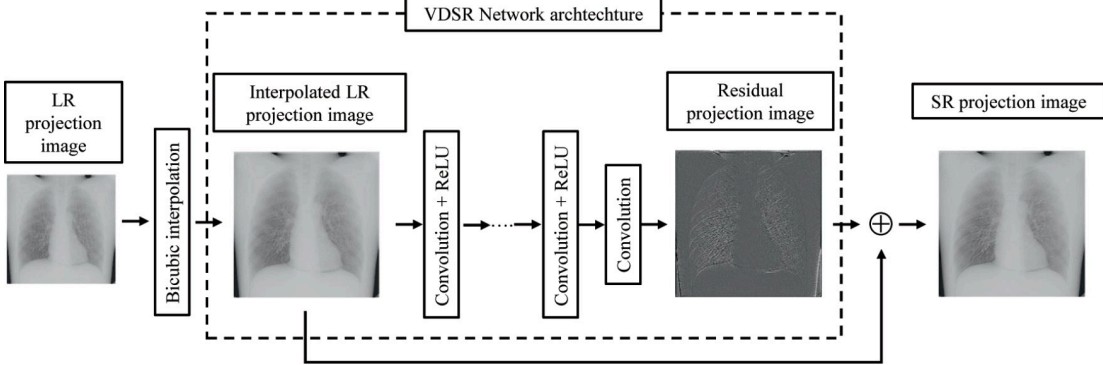

**Fig 4. Network architectures.** Convolutional neural network for super-resolution chest digital tomosynthesis.

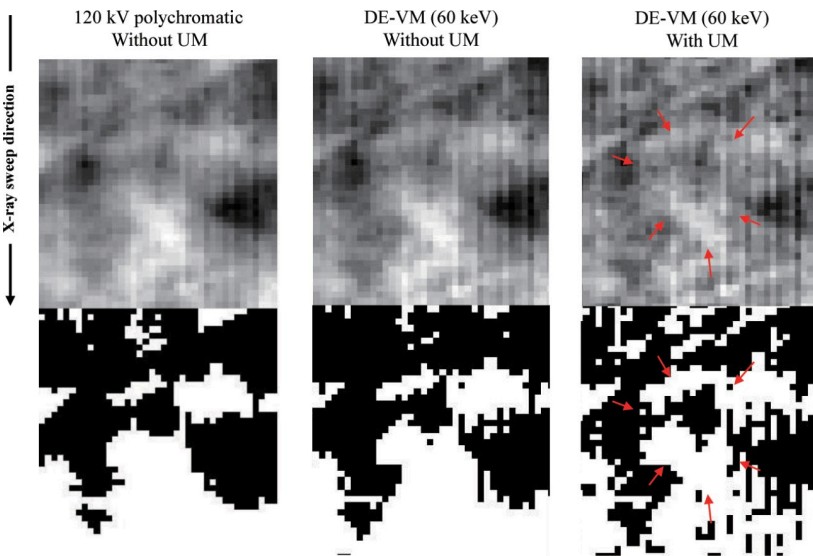

**Fig 5. Differences in simulated GGO nodule detection on polychromatic and Virtual Monochromatic (VM) projection-based images.** The visibility of the simulated GGO nodule in the projection image is improved by performing UM. The upper row shows the projection image (with/without UM), whereas the lower row shows the global image threshold using Otsu's method.

In a learning-based context, the training dataset $\{G_\tau, H_\tau\}_{\tau=1}^{\omega}$ is used for learning mapping from the LR images $H$ to the HR images $G$ ($G \in \Re^\Gamma$), resulting in the following equation:

$$\hat{G} = \text{argmin}_G \|H - ZG\|^2 \tag{3}$$

$$Z \in \Re^{\Gamma \times \Omega}, (\Gamma = \Omega)$$

Given a set of HR images $G_\tau$ ($G \in \Re^\Gamma$) and their corresponding LR images $H_\tau$ ($H \in \Re^\Omega$) with $\omega$ samples, the restoration operator $f$ can be estimated as follows:

$$\hat{f} = \arg\min_f \sum_{\tau=1}^{\omega} \frac{1}{2} \left\| G_\tau - f(S^\uparrow H_\tau; \delta) \right\|^2 \tag{4}$$

$S^\uparrow$: Upscaling interpolation operator
$\delta$: Trainable parameter
The network learns the residual errors between the output (HR image) and input (LR image) instead of the HR output.

The VDSR training workflow can be understood through the mini-batch SGDM method [49]. The hyperparameters of $\delta$ showed that the contrast of the resulting images may be affected by the mini-batch, epochs, and initial learning rate [44]. We determined the optimal parameters and applied them in the "Optimization parameters" section below.

$f$ is the mapping from the interpolated LR image to the HR image. VDSR with UM is used for estimating high-frequency components from the LR to HR and for estimating the mapping, also called as residual $r$, between HR and LR; r = $G$–($S^\uparrow H$). This method can be modeled through a skip connection in the network. Based on residual-based modeling, $r$ is assumed to be a function of $S^\uparrow H$. The final SR can be then obtained as follows:

$$G = S^\uparrow H + f(S^\uparrow H), \tag{5}$$

where *f* can be determined as follows:

$$\hat{f} = \arg\min_f \sum_{\tau=1}^{\omega} \left\| (G_\tau - (S^\uparrow H)_\tau) - f(S^\uparrow H)_\tau \right\|^2 \tag{6}$$

**First-Iterative Shrinkage–Thresholding Algorithm (FISTA).** SART–TV–FISTA algorithm consists of three steps: 1) SART update; 2) TV minimization; and 3) FISTA acceleration technique. The reconstruction process of SART can be given as follows:

$$s_j^{(k+1)} = s_j^k + \lambda \cdot \frac{\sum_{g_i \in G_\theta} \frac{g_i - \sum_{n=1}^{N} b_{in} \int_n^k s}{\sum_{n=1}^{N} b_{in}} \cdot b_{ij}}{\sum_{g_i \in G_\theta} b_{ij}}, \tag{7}$$

where $s_j$ refers to a voxel in the object function $s$, $g$ represents a single pixel in the DE–VM–SR projection data $G$, $\theta$ is the projection angle of view, $b_{ij}$ is an element in the matrix $B$ (discrete line integral of the system), $k$ is the iteration index, and $\lambda$ is the relaxation factor. TV minimization is the optimization problem motivated by the compressed sensing (CS) theory [50]. This technique has been applied to tomography [51] and can be given as follows:

$$\vec{s}^* = \arg\min \|s\|_{TV}, \tag{8}$$

$$|B \cdot \vec{s}^* - hr| < \delta, \tag{9}$$

$$\|s\|_{TV} = \sum_{i,j,k} \sqrt{(s_{i,j,k} - s_{i-1,j,k})^2 + (s_{i,j,k} - s_{i,j-1,k})^2 + (s_{i,j,k} - s_{i,j,k-1})^2}, \tag{10}$$

where $\vec{s}^*$ is the volume image to be recreated, $\delta$ relates to the level of data inconsistency tolerance, and $\|s\|_{TV}$ is the $l_1-norm$ of the image gradient magnitude used as the cost function. $s_{i,j,k}$ is the value of the voxel with index $(i,j,k)$. The standard steepest descent technique was employed to solve Eq (8). The steepest gradient method based on an iterative scheme can be used to minimize the TV objective function. In this method, the image is updated at each iteration $n$ ($n = 1$ *to* 20) as follows:

$$s^{(n+1)} = s^{(n)} - \beta \cdot d_p \cdot \frac{ds^n}{|ds^n|} \tag{11}$$

$$d_p = |s - s_0|$$
$$ds^n = \nabla s \|s\|_{TV.}$$

The value of $d_p$ can be initially computed and is observed to be dependent on the values of $s_0$ and $s$ corresponding to the image values before and after the SART update step, respectively. $\nabla$ represents the gradient operator, and $\beta$ is a parameter used to control the effect of regularization. The regularization parameter $\beta$ was set to 1e-7 for maximizing the simulated nodule contrast. The FISTA acceleration step is used for convergence and can be given as follows:

$$s^{(m+1)} = s^m + \left( \frac{t^m - 1}{t^{(m+1)}} \right) (s^m - s^{(m-1)}), \tag{12}$$

where the parameter $t$ is updated during each iteration as follows:

$$t^{(m+1)} = \frac{1 + \sqrt{1 + 4(t^m)^2}}{2} \tag{13}$$

$t = 1$ is the initial given value, and $m$ is the index of the global iteration step.

**Bilateral Filtering (BF).**   BF performs smoothing while preserving edges, such as contours, and is applied to enhance image quality by reducing the noise in medical images [26]. The objective of using BF in this algorithm was to suppress and adjust the increase in high-frequency components (especially noise) after reconstruction. The BF of the identified in-focus plane image can be defined as follows:

$$bf(i,j) = \frac{\sum\limits_{\gamma=-\eta}^{\eta}\sum\limits_{\varepsilon=-\eta}^{\eta} s(i+\varepsilon, j+\gamma) \exp\left(-\frac{\varepsilon^2 + \gamma^2}{2\sigma_d{}^2}\right) \exp\left(-\frac{(s(i,j) - s(i+\varepsilon, j+\gamma))^2}{2\sigma_R{}^2}\right)}{\sum\limits_{\gamma=-\eta}^{\eta}\sum\limits_{\varepsilon=-\eta}^{\eta} \exp\left(-\frac{\varepsilon^2 + \gamma^2}{2\sigma_d{}^2}\right) \exp\left(-\frac{(s(i,j) - s(i+\varepsilon, j+\gamma))^2}{2\sigma_R{}^2}\right)}, \tag{14}$$

where $\eta$ is a set of neighborhood points around the pixel, $\sigma_d$ is the standard deviation of the domain filter, $\sigma_R$ is the standard deviation of the range filter, and $bf$ is applied in VDSR of the final DE–VM–VDSR image. The parameters are a set of $\eta$ [$\eta = 2\cdot(2\cdot\sigma_d)+1$: $\sigma_d = 1$; 5, $\sigma_d = 2$; 9, and $\sigma_d = 3$; 13] and $\sigma_R$ ($\sigma_R = 0.01$ $[0, 1.0]^2{}_{img\_scale\_diff}$; [, ]$_{img\_scale\_diff}$; scale range of image, $\sigma_R = 0.01$). $\sigma_d$ is the parameter that must be considered when studying the effect of BF on contrast. We assessed the optimal parameter $\sigma_d$ and applied the parameters in the "Optimization parameters" section.

## Evaluation

### Optimization parameters of VM energy, initial learning rate, mini-batch size, epochs, number of iterations ($k$, $m$) for IR, and standard deviation of domain filter ($\sigma_d$)

As in previous chest DT studies, the FBP (kernel: Ramachandran–Laksminarayanan) and SART (polychromatic 120kV acquisition) were used [27]. The number of iterations must be optimized because the characteristics of the IR image are dependent on the number of iterations. First, the iteration number is optimized in polychromatic SART and SART–TV–FISTA using the root-mean square error (RMSE). The RMSE of the identified in-focus plane image (cross-sectional image of the center of rotation) can be given as follows:

$$RMSE = \sqrt{\frac{1}{\tau\varsigma} \sum_{i=0}^{\tau-1} \sum_{j=0}^{\varsigma-1} \left[K_{ref}(i,j) - V_{obj}(i,j)\right]^2}, \tag{15}$$

where $K_{ref}(i,j)$ is the $(i,j)$th entry of the current iteration image and $V_{obj}(i,j)$ is the $(i,j)$th entry of the previous iteration image of each algorithm.

Then, the optimal iteration number of the polychromatic (120 kV) SART–TV–FISTA image was applied to monochromatic SART–TV–FISTA (DE–VM–SART–TV–FISTA) and DE–VM–VDSR. The optimizations of DE–VM energy, initial learning rate, mini-batch size, epochs, and standard deviation of the domain filter were evaluated based on the signal-difference-to-noise ratio (SDNR) in case of the in-focus plane image. The SDNR of the identified

in-focus plane image can be given as follows:

$$SDNR = \frac{|\Lambda_{GGO} - \Lambda_{BG}|}{\sigma_{BG}}, \tag{16}$$

where $\Lambda_{GGO}$ and $\Lambda_{BG}$ are the mean pixel intensities inside the simulated GGO nodule and background (BG) fields, respectively, and $\sigma_{BG}$ is the standard deviation of multiple regions of interest (ROIs) (radius: 9 pixels) in the BG field. The background signal were represented by the multiple circular ROIs inside the lung field (Fig 6A). The DT system-derived real projection data were used to achieve reconstruction. MATLAB (MathWorks, Natick, MA) was used for reconstructing and processing images. Optimization was evaluated based on the RMSE and SDNR. The lowest RMSE for iteration number, the highest SDNR for VM energy, initial learning rate, mini-batch size, epochs, and standard deviation of domain filter were selected as the optimum parameters.

## Evaluation of image quality

The SDNRs in case of DE–VM–VDSR with and without BF and the conventional algorithms (polychromatic FBP, SART, SART–TV–FISTA, DE–VM–SART–TV–FISTA with reconstruction from the original projections) were compared to assess the increase in simulated nodule contrast on each in-focus plane image. We further ascertained the ripple artifacts, spatial resolution, and noise. Gumbel distributions [52] are statistical models that can be used for determining the influence of ripple artifacts. The radial modulation transfer function (radial MTF) [53] shows the spatial resolution of the simulated GGO nodule on the features of the images in the in-focus plane. The noise power spectrum (NPS) [54] is the noise of a uniform object on a feature of an in-focus plane image. The characteristics of DE–VM–VDSR with and without BF and the conventional algorithms were evaluated based on the contrast, ripple artifact reduction, spatial resolution, and noise. DE–VM–VDSR with and without BF was evaluated using the optimized parameters generated based on the application image.

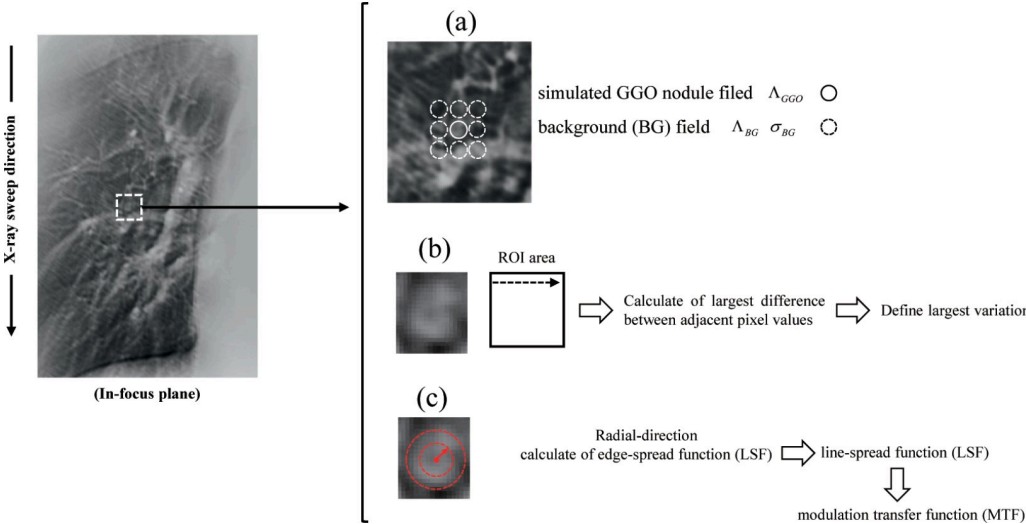

**Fig 6. Assessing improvements in image quality using the Signal-Difference-to-Noise Ratio (SDNR), a statistical model with a Gumbel distribution, and the radial Modulation Transfer Function (radial MTF) of the selected features.** (a) The in-focus plane image shows the simulated ground-glass opacity (GGO) nodule and background areas of the SDNR, (b) measurements and the ripple artifact with simulated GGO nodule areas of the Gumbel analysis, and (c) measurements of the simulated GGO nodule areas of the radial MTF.

## Statistical model with a Gumbel distribution

Based on Gumbel distribution, the streak artifacts of the high-frequency components found in cross-sectional images can be quantitatively analyzed [46, 52]. Therefore, we selected to use this statistical model. The analysis procedure is shown below.

1. A rectangular window with a width of 24 pixels and a length (X-ray sweep direction) of 24 pixels was placed on each in-focus plane image nearly perpendicular to multiple streak artifacts, as shown in Fig 6B.

2. The parallel-line profiles of the pixel values at 1-pixel intervals (as shown by the arrow in Fig 6B) resulted in a total of 23 parallel-line pixel-value profiles (each sampling size: 23).

3. The pixel-value profiles were graphed, and the maximal variations between adjacent pixel values were determined and analyzed based on the Gumbel distribution.

4. The cumulative probability function was measured using the symmetry rank method [cumulative probability $Q(x_{(\partial)}) = \frac{(\partial - 0.5)}{\ell}$, (for $\partial = 1, ...\ell$); $\ell$ is the sampling size] with order statistics.

5. Based on the generated probability diagram, the maximal difference between adjacent pixels showed linearity with a cumulative probability; thus, the Gumbel characteristic could be calculated based on the maximal difference between neighboring pixels.

Finally, to evaluate linearity, Pearson's correlation coefficient was determined and examined [probability (*P*) values < 0.01] using IBM SPSS Statistics for Windows (version 24.0; IBM Corp., Armonk, NY).

## Radial MTF

Radial MTF [53] considers that the edge of a circular object is a combination of all the edges in a line extending in the radial direction from the center of the circle and acquires several profiles in the radial direction from the center of the object. The edge-spread function is obtained by averaging the pixel-value profile of the edge image, the line-spread function is obtained from the difference, and MTF is obtained based on the one-dimensional fast Fourier transform. The spatial resolution of the simulated GGO nodule was calculated in the radial direction, including the BG field from the center of the nodule as the evaluation area (radius: nine pixels, area: 254 pixels) (Fig 6C).

## Noise Power Spectrum (NPS)

Based on NPS, a uniform water phantom was acquired (vertical: 300 mm, horizontal: 200 mm, longitudinal: 450 mm) and analyzed using the two-dimensional Fourier analysis method [54] by considering an in-focus plane. The ROI (64 × 64 pixels) was set to 8 pixels in the vertical direction and the horizontal direction in the area near the center of the field of view (256 × 256 pixels); 64 ROIs were set, and the average value was considered to be the NPS. Trend correction was achieved at each ROI through polynomial approximation (quadratic). The exposure parameters were similar to those obtained when the chest phantom N1 was acquired.

# Results

## Optimization parameters

After measuring the RMSE of each iteration, the optimal number of iterations (*k*, *m*) in IR for SART–TV–FISTA converged to 30 and SART converged to 24 (Fig 7A). Therefore, the

number of iterations was set to 30 for SART–TV–FISTA and 24 for SART. The energy in DE–VM processing was set to 60 keV because it resulted in the highest SDNR in VM energy optimization of DE–VM–SART–TV–FISTA (Fig 7B). Next, the SDNR was measured with epochs of 5 and 10 to 80 (interval: 10) to verify the optimization with different initial learning rates (0.1, 0.01, and 0.001) and mini-batch sizes of 64, 128, 256, and 512. According to the optimization verification conducted by Kim et al. [44] (evaluated by peak signal-to-noise ratio (PSNR), epoch changes to around 40 but tends to be constant (converge) after 40. With reference to these results, the highest values of epoch and mini-batch indicating a tendency of convergence were selected as the optimization values. The SDNR value was the highest for epochs of 70 and a mini-batch size of 128 with an initial learning rate of 0.001 (Fig 8A–8C). Therefore, learning was performed by setting the mini-batch size to 128 and epochs to 70, with an initial learning rate of 0.001 (hyper-parameter). SDNR was measured by changing $\sigma_d$ to 1, 2, and 3 to optimize the domain filter ($\sigma_d$) standard deviation of bilateral filtering, and the highest SDNR was 1 and 2 (Fig 8D). The increase in $\sigma_d$ is affected by blurring with smoothing [26]. Therefore, $\sigma_d$ was set to 1. Using the results of optimization verification, DE–VM–VDSR images were generated by setting the number of iterations ($k$, $m$) to 30, mini-batch size to 128, epochs to 70, and standard deviation of the domain filter ($\sigma_d$) to 1; the simulated nodule contrast (SDNR), ripple artifact

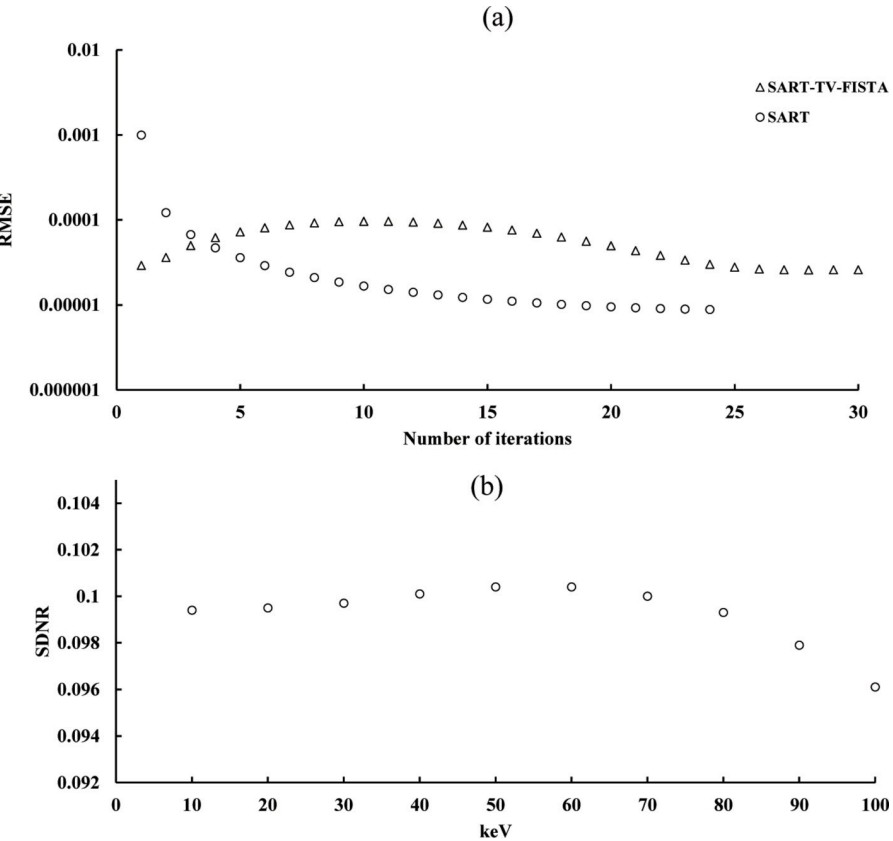

**Fig 7. Optimization results for parameter [number of iterations for Iterative Reconstruction (IR) and Dual-Energy Virtual Monochromatic (DE–VM) energy] determination.** (a) The root-mean square error (RMSE) with respect to number of iterations are shown for each polychromatic IR algorithm. (b) The signal-difference-to-noise ratios (SDNR) resulting from differences in the DE–VM energy in the simultaneous algebraic reconstruction technique (SART) total-variation (TV) first-iterative shrinkage–thresholding algorithm (FISTA) [DE–VM–SART–TV–FISTA] are shown.

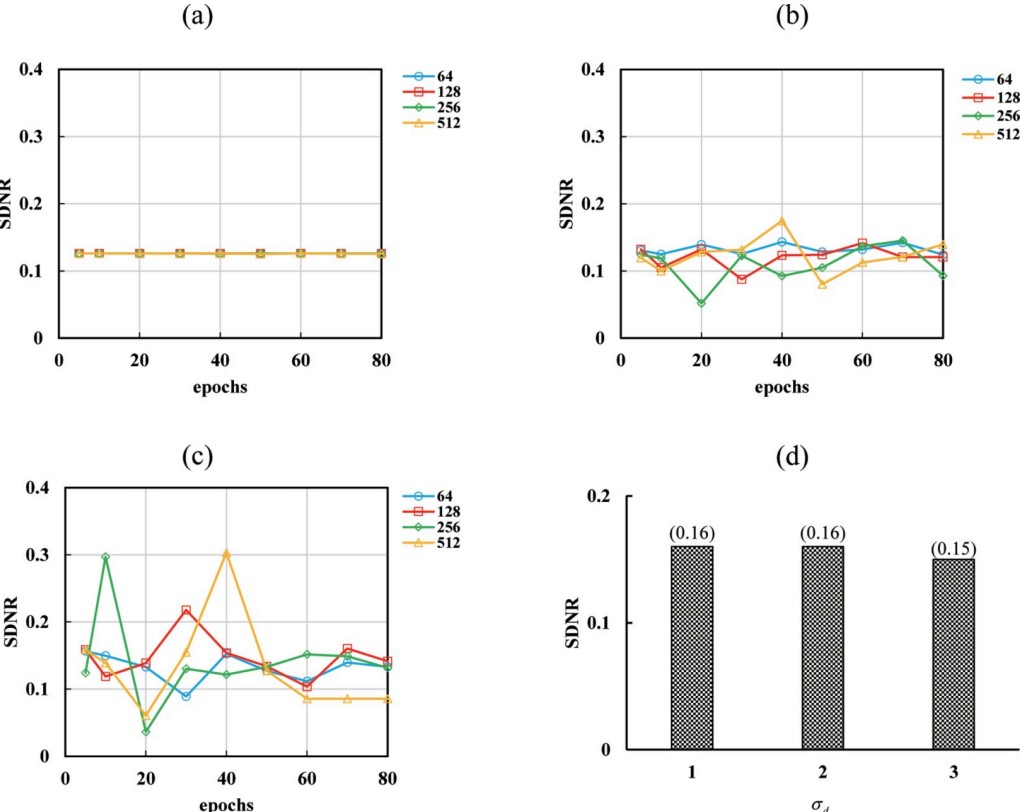

**Fig 8. Optimization results for parameter [mini-batch, epochs, and standard deviation of domain filter $\sigma_d$] determination.** The signal-difference-to-noise ratio (SDNR) resulting from differences in the initial learning rate [(a): 0.1, (b): 0.01, and (c): 0.001], mini-batch, and epochs in the DE–VM with the very-deep super-resolution (VDSR) reconstruction algorithm (DE–VM–VDSR) are shown. (d) The SDNRs resulting from differences in $\sigma_d$ in DE–VM–VDSR are shown. From the results (Figs 7 and 8) of optimization verification, DE–VM–VDSR images were generated by setting the number of iterations to 30, initial learning rate to 0.001, mini-batch size to 128, epochs to 70, and $\sigma_d$ to 1.

(Gumbel distribution), spatial resolution (radial MTF), and noise (NPS) were evaluated and compared with those of the images obtained using conventional algorithms.

## Image quality

Fig 9 shows the reconstructed images of the chest phantom N1 with a simulated GGO nodule using DE–VM–VDSR with and without BF or the conventional reconstruction algorithms. Remarkably, the DT images generated by DE–VM–VDSR with BF algorithm showed decreased ripple artifacts and increased simulated nodule contrast. Further, the edges on the simulated nodule were sharpened and more clearly identified. In contrast, the images produced with polychromatic FBP exhibited more noise and ripple artifacts, and polychromatic SART, SART–TV–FISTA, and DE–VM–SART–TV–FISTA exhibited reduced sharpness and contrast in the simulated nodule.

**SDNR.** Fig 10 shows the SDNR results for the ROI set on the chest NI phantom. DE–VM–VDSR with BF yielded the highest simulated GGO nodule contrast regardless of the usage of the conventional algorithm. The simulated GGO nodule contrast was dependent on the type of reconstruction algorithm in case of the polychromatic imaging algorithms, FBP, SART, and SART–TV–FISTA, and DE–VM–SART–TV–FISTA.

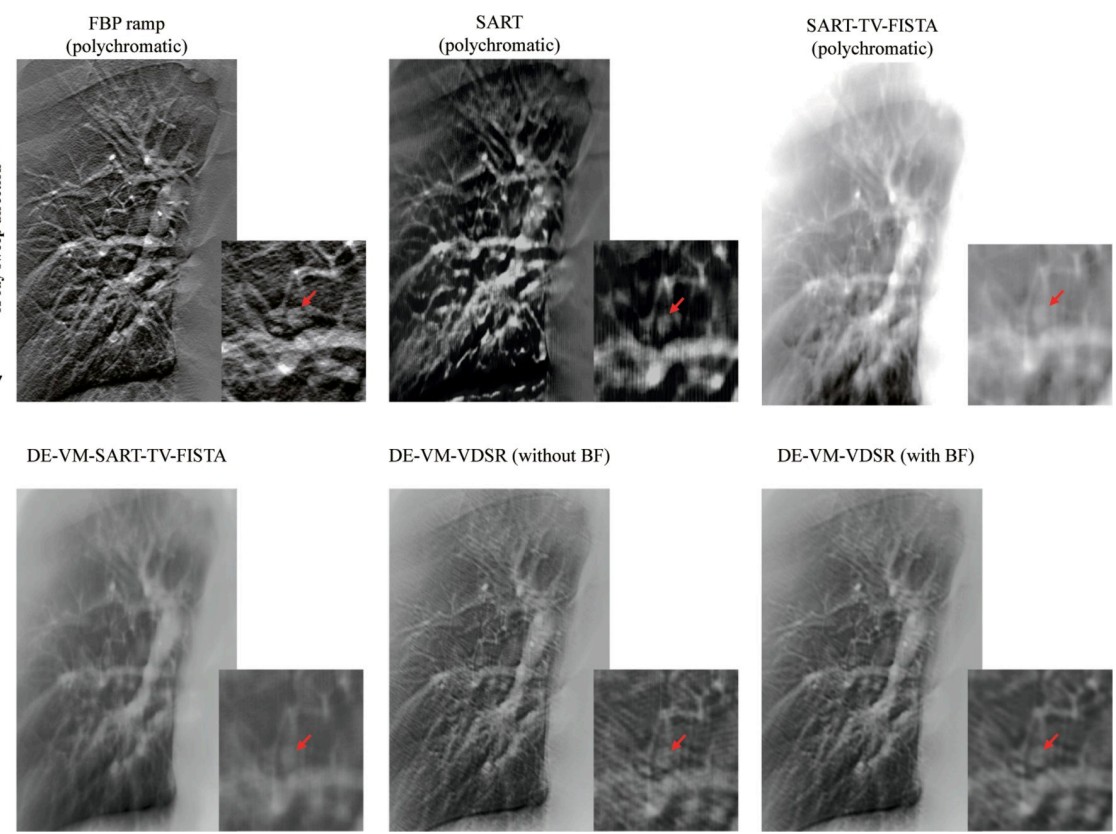

**Fig 9. Comparisons among the Dual-Energy (DE) Virtual Monochromatic (VM) with Very-Deep Super-Resolution (VDSR) reconstruction algorithm (DE–VM–VDSR) with and without BF and conventional reconstruction algorithms [DE–VM–VDSR with and without BF (showing window: 0–0.23), FBP (kernel: Ramp; 0–0.4), SART (120 kV; 0–0.02), SART–TV–FISTA (120 kV; 0–0.4), and DE–VM–SART–TV–FISTA (60 keV; 0–0.23)] in the in-focus plane.** The window of the chest phantom N1 with lung field was varied to compare the contrast and background gray levels. For each corresponding set, the VM (DE–VM–VDSR and DE–VM–SART–TV–FISTA) images are displayed at the same window width and level, whereas the polychromatic FBP and IR images have larger window widths because the backgrounds are less flattened. The X-ray source is moved vertically along the image. Abbreviations: FBP = filtered backprojection, SART = simultaneous algebraic reconstruction technique, TV–FISTA = total-variation first-iterative shrinkage–thresholding algorithm, IR = iterative reconstruction, BF = bilateral filtering.

**Gumbel distribution.** Fig 11 presents a Gumbel plot of the relations between the maximal variations and predicted cumulative probabilities (DE–VM–VDSR with BF = $0.0087 \pm 0.0006$, DE–VM–VDSR without BF = $0.0142 \pm 0.0010$, FBP = $0.0198 \pm 0.0013$, SART = $0.0142 \pm 0.0009$, SART–TV–FISTA = $0.0099 \pm 0.0005$, SART–TV–FISTA with VM = $0.0102 \pm 0.0005$). Here, the largest variations are linearly distributed (DE–VM–VDSR with BF, $r = 0.908$ [$P < 0.01$]; DE–VM–VDSR without BF, $r = 0.968$ [$P < 0.01$]; polychromatic FBP, $r = 0.971$ [$P < 0.01$]; polychromatic SART, $r = 0.963$ [$P < 0.01$]; polychromatic SART–TV–FISTA, $r = 0.943$ [$P < 0.01$]; and DE–VM–SART–TV–FISTA, $r = 0.938$ [$P < 0.01$]). These observations validated the use of Gumbel distribution as an acceptable statistical model for defining the largest variations in differences between the nearby pixel-value profiles. Additionally, based on analysis of the maximal variations in the Gumbel plot, DE–VM–VDSR with BF yielded the lowest high-frequency ripple artifacts. The polychromatic SART–TV–FISTA algorithms exhibited fewer high-frequency ripple artifacts than DE–VM–SART–TV–FISTA, whereas the polychromatic FBP algorithm distribution differed from other algorithms.

**MTF.** The spatial resolution of the simulated GGO nodule region is shown in Fig 12. DE–VM–VDSR without BF exhibited improvement in the low-frequency region (0.2–0.3 cycles/

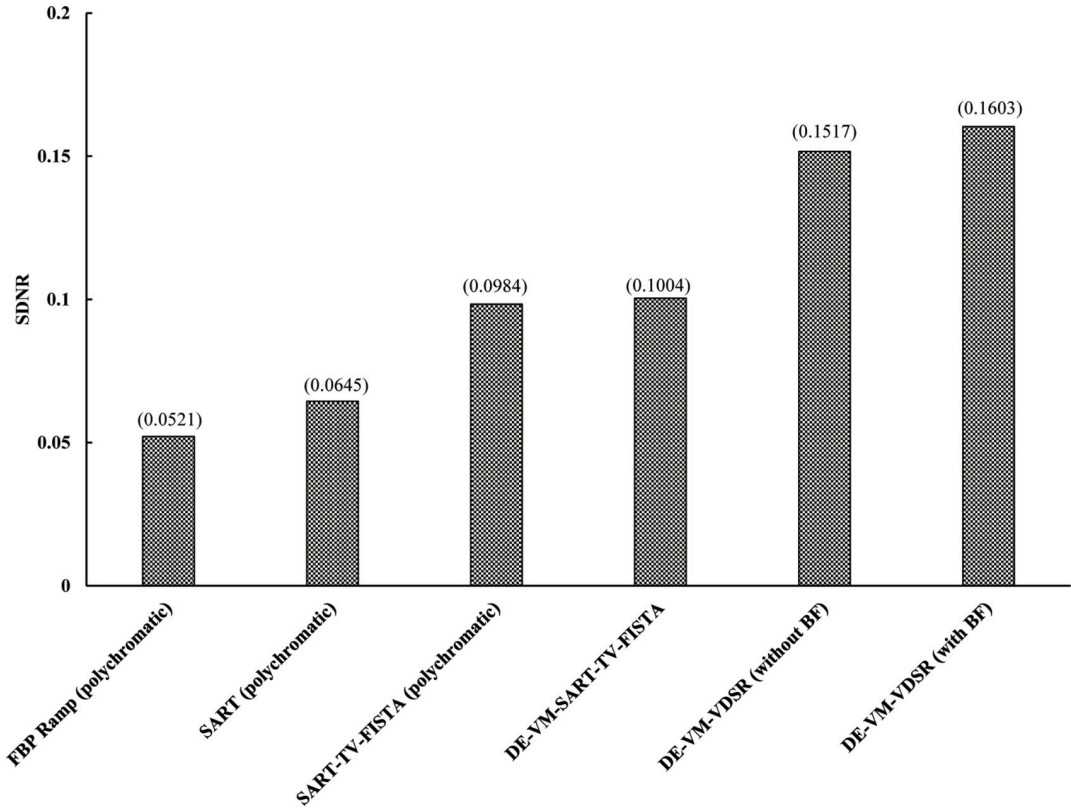

**Fig 10. Plots of the SDNR versus each algorithm from the in-focus plane.** Comparisons of the signal-difference-to-noise ratio (SDNR) of in-focus plane images obtained via the dual-energy (DE) virtual monochromatic (VM) with the very-deep super-resolution (VDSR) reconstruction algorithm (DE–VM–VDSR) with and without BF and the conventional reconstruction algorithms.

mm), whereas FBP improved in the high-frequency region (0.7–1.0 cycles/mm). When comparing DE–VM–VDSR with and without BF, the spatial resolution without BF was better. The spatial resolution tended to decrease for polychromatic SART–TV–FISTA and DE–VM–SART–TV–FISTA. Among the conventional algorithms, polychromatic SART–TV–FISTA showed a similar spatial resolution to that exhibited by DE–VM–SART–TV–FISTA.

**NPS.** Fig 13 presents the noise characteristics of the reconstructed image in a uniform water phantom. DE–VM–VDSR with BF showed the lowest noise characteristics. In particular, the low noise in the low-frequency region (~0.8 cycles/mm). When comparing DE–VM–VDSR with and without BF, BF resulted in lower noise. The noise characteristic tended to increase for FBP and polychromatic SART.

## Discussion

The newly established DE–VM–VDSR algorithm, in which DE–VM and VDSR are combined with UM for projection-space processing with SART–TV–FISTA prior to reconstruction with BF, achieved adequate overall functioning in this study. This hybrid algorithm resulted in an improved image quality with respect to the simulated GGO nodule present in the chest phantom. Furthermore, this algorithm effectively reduced the ripple artifacts from the images, specifically at large distances from the ribs. DE–VM–VDSR with BF was useful for enhancing the pulmonary GGO nodule contrast. In summary, this algorithm may be a novel alternative with promising applications for chest imaging because images considerably superior to those

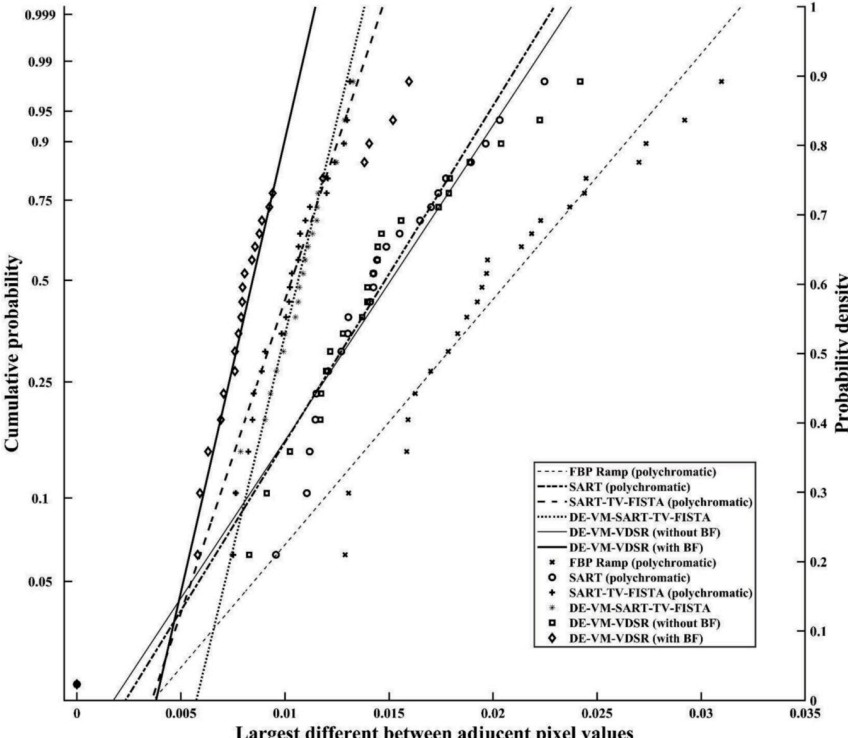

**Fig 11. The largest variations extracted from 23 pixel-value profiles are plotted.** The relatively large variations in pixel values were attributed to high-frequency ripple artifacts.

obtained using conventional algorithms can be achieved with respect to the contrast, ripple artifacts, and noise reduction. Based on the chest imaging conditions and required characteristics of the final images, DE–VM–VDSR with BF provides flexibility with respect to the selection of imaging parameters that may be helpful to users.

The VM method can be used to produce material-selective images devoid of beam-hardening artifacts. Because the mass density of each basis material at each point in the image is dependent on the linear attenuation coefficients of the basis materials at a nominal photon energy, the beam-hardening artifacts can result in erroneous mass densities of the basis materials [46]. Thus, our new DE–VM–VDSR with BF, which is created by combining VM-VDSR processing and SART–TV–FISTA after reconstruction of the BF algorithms, could effectively enhance contrast and reduce ripple artifacts (Figs 10 and 11).

VDSR is a training strategy based on which an image of any scale can be reconstructed. Thus, a hybrid training strategy is helpful for SR images. A well-trained network can process images of all scales and reconstruct the input image to any size. VDSR recreates the residual image, making it simpler to examine the difference between LR and HR [55]. The edge and texture generated using VDSR with UM are sharp. Residual learning can estimate the residuals between LR and SR images so that the final SR images can be obtained using LR images. Images obtained via DE–VM–VDSR without BF can be visualized by this feature in the radial MTF plot (Fig 12). However, the ripple artifacts and noise increased as the spatial resolution improved. BF is a useful tool for preserving the enhanced edge information and reducing ripple artifacts and noise.

DE–VM–VDSR with BF showed good characteristics with same results regarding ripple artifacts and noise reduction. In particular, it can effectively reduce noise in the high-frequency region and helped improve contrast. These effects are thought to have resulted from the

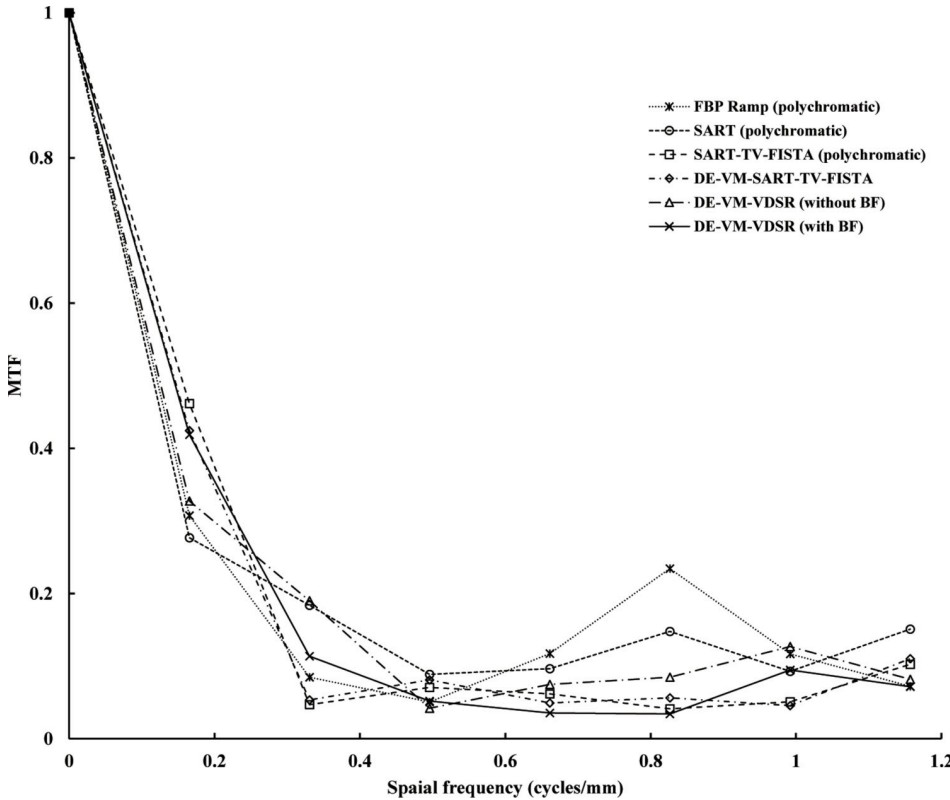

**Fig 12. Plots of the radial Modulation Transfer Function (radial MTF) versus spatial frequency (cycles/mm) from the in-focus plane for the Dual-Energy (DE) Virtual Monochromatic (VM) with Very-Deep Super-Resolution (VDSR) reconstruction algorithm (DE–VM–VDSR) with and without BF and conventional reconstruction algorithms.** The spatial frequency of the simulated ground-glass opacity (GGO) nodule area indicates the simulated GGO nodule and background areas of the radial MTF measurements. The plots are of the area of the GGO nodule and background regions.

improvement of contrast and resolution by the combination of DE–VM and VDSR with UM and from the effective ripple artifacts and noise reduction in the state of edge preservation by BF (Figs 10, 11 and 13).

Through CNN-based approaches, the upscaling process can use the transposed convolution (so-called fractionally strided convolutional) layers [56, 57] or sub-pixel layers [58]. However, the weights of these trained networks are set for a specific scale factor [59]. This is a restrictive feature of CNN-based SR for DT projection data because a fixed upscaling factor is not ideal in this scenario. This is because high-frequency image content cannot be retrieved from the LR image when using fixed upscaling. We have presented a multiscale CNN-based SR method for three-dimensional CDT that can learn multiple scales by training multiple scale factors employing an independent upscaling method, for example, bicubic interpolation.

The following two factors are critical to ensure outstanding functioning of the CNN-based SR algorithm: selecting the procedures for evaluating the enhancements in image quality (contrast and spatial resolution) and generating adequate training data. The choice of an appropriate image-quality-enhancing method dictates whether sufficient data are available for the CNN-based SR to differentiate the simulated GGO nodules from normal structures. The training data for CNN-based SR comprises as many pulmonary GGO nodule types as possible.

In FISTA, the initial value of the next iteration can be evaluated by linearly combining the results of the two earlier iterations. CDT image reconstruction is normally scarce in some local

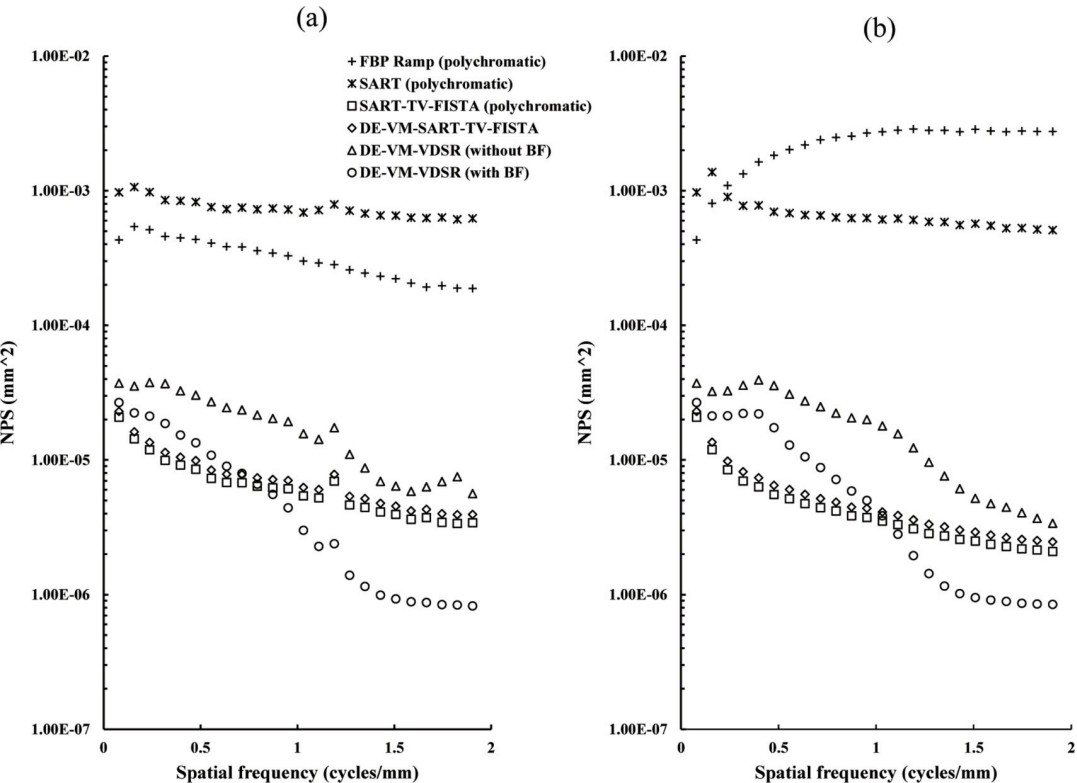

**Fig 13. Plots of the Noise Power Spectrum (NPS) versus spatial frequency (cycles/mm) from the in-focus plane for the Dual-Energy (DE) Virtual Monochromatic (VM) with Very-Deep Super-Resolution (VDSR) reconstruction algorithm (DE–VM–VDSR) with and without BF and conventional reconstruction algorithms.** The NPS was obtained via two-dimensional Fourier analysis of the central field of view from the reconstructed image of the uniform water phantom. (a) Horizontal directions and (b) vertical directions.

high-frequency regions because the GGO nodules are dispersed inside specific lung fields. This suggests that a combination of regularizations that increase the effect of the conditions of sparsity and smoothness could enhance the recreation of CDT. The most basic sparse index is the $L_0$ norm, which is the number of nonzero elements in all vectors. However, the normal inverse problem of $L_0$ is difficult to solve, at least in the undetermined case [60]. In contrast, the $L_1$ norm is a convex relaxation of the $L_0$ norm and frequently employed to improve the sparsity of images, specifically in the field of CS [50, 61]. The optimization difficulties can be resolved using many algorithms, such as gradient projection [62], iterative shrinkage–thresholding [63], and FISTA [29, 30]. FISTA is ideal because it can assess fast and precise $L_1$ solutions that can solve the reconstruction problem and improve the sparsity of CDT images.

TV minimization assumes that a true image is comparatively uniform and piecewise. Because TV is defined as the sum of the first-order derivative magnitudes [64], noise and artifacts can be observed as valleys, and peaks have relatively larger TV values. Thus, TV is restricted to clinical CT. Through the combination of the CNN-based SR algorithm and FISTA–TV, nonlinear regression can be learned. Further, prior knowledge can be effectively used in future pulmonary GGO nodules and CDT images.

Preferably, the structures in each plane of interest must be clearly displayed in the corresponding plane of DT reconstruction, whereas those present outside the plane must be rendered invisible [65]. Fundamentally, the restricted angular range of the DT image-acquisition geometry restricts spatial resolution in the dimension perpendicular to the detector plane [65].

Consequently, out-of-plane structures cannot be totally reduced from the reconstruction plane; therefore, they are always present. However, majority of these structures are invisible because different low-amplitude structures from projections overlap in the reconstruction plane and cause image blurring. On the contrary, out-of-plane structures from high-attenuation features cannot be lowered and show up as replicates in each reconstruction plane except in the plane in which the actual high-attenuation feature (dorsal lib) can be observed. At one projection angle, ghosting features (ripple artifacts) are spread along the line between the actual feature and X-ray source. DE–VM–VDSR with BF has been beneficial for reducing the ripple artifacts but may result in the elimination of the needed clinical features.

Because CDT acquisition requires an imaging time of 6.4 s, it may be difficult to hold breath depending on the patient's condition when assuming clinical use. Currently, patients whose condition is stable will be considered for examination. We hope that the acquisition time can be reduced by improving the specifications of the acquisition system (detector sensitivity, hardware control, etc.).

Our DE–VM–VDSR with BF has some limitations. First, even though the three-material decomposition method was selected to estimate the numerical stability, further perfections are required in this area. The precision of three-material decomposition is restricted on current DT systems, specifically for elements with low fractions [66]. Second, the indicated learning model is proposed for a specific type of phantom with simulated pulmonary nodules and may not work efficiently when the trained network is adjusted to projection-data correction from entirely different scanning geometries. Future studies are required to design a learning model that can be employed in general cases. Third, DE acquisition is a mechanical constraint, and not many devices can quickly switch between tube voltages during a single acquisition. Finally, HR projections in training the VDSR network is the original CDT acquisitions. Because all the LR projections are artificially downsampled, obtaining LR projections is an important step in the proposed workflow to reduce the imaging noise and processing time.

## Conclusion

In this chest phantom study, we compared the newly developed DE–VM–VDSR with BF and various conventional DT reconstruction algorithms without and with VM processing. The features of the new DE–VM–VDSR with BF are implemented by combining preprocessing (VM and VDSR with UM) and postprocessing (BF). The performance of DE–VM–VDSR with BF was evaluated for contrast (SDNR), ripple artifacts (Gumbel distribution), resolution (radial MTF), and noise (NPS). Particularly, our DE–VM–VDSR with BF increased the contrast, reduced high-frequency ripple artifacts, and generated relatively better noise reduction results when compared with those of conventional reconstruction algorithms.

## Supporting information

**S1 File.**
(ZIP)

## Acknowledgments

We wish to thank Mr. Kazuaki Suwa and Yuuki Watanabe at Department of Radiology Dokkyo Medical University Koshigaya Hospital for support on experiment.

## Author Contributions

**Conceptualization:** Tsutomu Gomi.

**Data curation:** Tsutomu Gomi.

**Formal analysis:** Tsutomu Gomi.

**Funding acquisition:** Tsutomu Gomi.

**Investigation:** Tsutomu Gomi.

**Methodology:** Tsutomu Gomi.

**Project administration:** Tsutomu Gomi.

**Resources:** Tsutomu Gomi.

**Software:** Hidetake Hara.

**Supervision:** Tsutomu Gomi.

**Validation:** Yusuke Watanabe.

**Visualization:** Shinya Mizukami.

**Writing – original draft:** Tsutomu Gomi.

**Writing – review & editing:** Tsutomu Gomi.

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
