## [Decision Letter · Decision Letter 0]

28 Aug 2020

PONE-D-20-19441

Digital chest tomosynthesis image quality improvement employing projection-based dual-energy virtual monochromatic convolutional neural network with super-resolution

PLOS ONE

Dear Dr. Gomi,

Thank you for submitting your manuscript to PLOS ONE. After careful consideration, we feel that it has merit but does not fully meet PLOS ONE’s publication criteria as it currently stands. Therefore, we invite you to submit a revised version of the manuscript that addresses the points raised during the review process.

When you submit your revised version, please highlight the novelty of this work and respond to the reviewers' comments.

We look forward to receiving your revised manuscript.

Kind regards,

Jeonghwan Gwak, PhD

Academic Editor

PLOS ONE

Journal Requirements:

2. Please amend your Data availability statement to clarify how other researchers may obtain the datasets and images used in this study. We note for instance that no data has been provided in the Supplemental information.

PLOS defines a study's minimal data set as the underlying data used to reach the conclusions drawn in the manuscript and any additional data required to replicate the reported study findings in their entirety. All PLOS journals require that the minimal data set be made fully available. For more information about our data policy, please see http://journals.plos.org/plosone/s/data-availability.

'This study was supported by a grant from Kitasato University School of Allied Health Sciences (Grant-in-Aid for Research Project, No. 2020-1006).'

'The author(s) received no specific funding for this work.'

Reviewers' comments:

Reviewer's Responses to Questions

**Comments to the Author**

1. Is the manuscript technically sound, and do the data support the conclusions?

Reviewer #1: Partly

Reviewer #2: Partly

2. Has the statistical analysis been performed appropriately and rigorously? 

Reviewer #1: Yes

Reviewer #2: Yes

3. Have the authors made all data underlying the findings in their manuscript fully available?

Reviewer #1: Yes

Reviewer #2: No

4. Is the manuscript presented in an intelligible fashion and written in standard English?

Reviewer #1: No

Reviewer #2: No

5. Review Comments to the Author

Reviewer #1: The authors propose a complicate, but novel, algorithm, which combines VDSR, DE-VM-SART-TV-FISTA and bilateral filtering, to improve CNR and reduce ripple artifacts for dual-energy chest tomosynthesis. Overall, this seems a good and promising work. However, some problems about method, results and presentation of the manuscript should be addressed before it fits the standard to be published.

Some major comments are given below. Please see the attached file to see the minor comments made on the manuscript PDF file.

* Training dataset:

It is not clear how the HR projections are obtained. If the authors used the original scans as the HR projections in the training, then what’s the logic to use VDSR to enhance image resolution? Our aim is to enhance the projections and let them have higher resolution, but you cannot get a good result unless you know the target. The authors claimed that using of multiscale and UM in the VDSR, maybe more explicit explanations and clear results are needed.

* Result comparison:

It is not clear if the DE-VM-SART-TV-FISTA contains the bilateral filtering step. It would be better if the authors could give comparisons with and without the VDSR and BF steps, respectively. Then the advantages of each of these two steps could be clearly shown. Also, the authors could show comparison of the projections with and without VDSR to demonstrate how the VDSR performs.

* Result problems:

a) The authors concluded that 280keV gives the highest contrast, which the reviewer highly doubts about. Please extend your exploration to lower energy to at least 10keV.

b) The optimal point in Fig. 6(c) for parameter optimization is an outlier. In the reviewer’s opinion, it should be double checked and confirmed with a line draw. The common practice is that, to debug a neural network, a coarse random searching in the parameter space, followed by fine grid searching in a specific small area, including the learning rate, etc.

c) The MTF results in Fig. 10 seems to be wrong: the spatial frequency range of the x-axis seems wrong; at 0.05 c/mm, the algorithm with highest MTF is FBP. Please check method and calculation.

d) Suggestion: It could be better to do some analysis on the noise behavior, such as noise power spectrum inspection of the multiple algorithms. However, it’s the authors decision.

* Language and presentation:

a) The sections of Introduction and Methods should be extensively revised to better show the logic of this work, why the authors choose to use the strategy, as well as all the technique details of this work. Some descriptions in the Methods could be moved to the Introduction. It could be better if the Methods could be organized as general whole picture first and specific points later. Reading the current version feels easily get lost for the audience.

b) Many grammar problems can be seen in the manuscript.

Reviewer #2: The paper presents the digital tomosynthesis image quality improvement using dual-energy virtual monochromatic very-deep super-resolution reconstruction algorithm.

1. Manuscript needs revision both at the level of concepts and write up.

2. There are too many grammatical mistakes in the manuscript. It should be corrected for reader’s better understand-ability.

3. References are written poorly, also authors may refer text not more than 5 years old because the technologies and research changes dynamically to day.

4. Introduction section does not have to be so long. It is better to use author's own words briefly.

5. Figures need much improvement in their qualities. The text is difficult to read in print form.

6. In my opinion it would be helpful if the authors provide some quantitative measures for accuracy in the abstract. for example, average dice, PSNR, or SDNR for all the testing data can be reported at the end of abstract.

7. Conclusion section requires revising to have better impact on the readers. Add 4-5 sentences to conclusion

8. Strongly recommend to improve the english quality to have better impact on the readers.

Below are few examples:

Introduction: page-3: i) " Even though, chest X-rays are less expensive with lower radiation exposure and easy availability, this technique, due to its low sensitivity, is inadequate for discovering lung nodules because." It should be corrected

ii) "It is not uncommon that during examinations of conventional chest radiographs, nearly 30% of such nodules may go undetected by chest radiologists with significant experience."

6. PLOS authors have the option to publish the peer review history of their article (what does this mean?). If published, this will include your full peer review and any attached files.

Reviewer #1: No

Reviewer #2: No

---

## [Author Response · Author response to Decision Letter 0]

22 Oct 2020

Inserted into the revised manuscript (PDF).

---

## [Decision Letter · Decision Letter 1]

1 Dec 2020

PONE-D-20-19441R1

Improved digital chest tomosynthesis image quality by use of a projection-based dual-energy virtual monochromatic convolutional neural network with super resolution

PLOS ONE

Dear Dr. Gomi,

Thank you for submitting your manuscript to PLOS ONE. After careful consideration, we feel that it has merit but does not fully meet PLOS ONE’s publication criteria as it currently stands. Therefore, we invite you to submit a revised version of the manuscript that addresses the points raised during the review process.

We look forward to receiving your revised manuscript.

Kind regards,

Jeonghwan Gwak, PhD

Academic Editor

PLOS ONE

Additional Editor Comments (if provided):

Please revise this manuscript to fully respond to reviewer 1's comments.

Reviewers' comments:

Reviewer's Responses to Questions

**Comments to the Author**

1. If the authors have adequately addressed your comments raised in a previous round of review and you feel that this manuscript is now acceptable for publication, you may indicate that here to bypass the “Comments to the Author” section, enter your conflict of interest statement in the “Confidential to Editor” section, and submit your "Accept" recommendation.

Reviewer #1: All comments have been addressed

Reviewer #2: All comments have been addressed

2. Is the manuscript technically sound, and do the data support the conclusions?

Reviewer #1: Yes

Reviewer #2: Yes

3. Has the statistical analysis been performed appropriately and rigorously? 

Reviewer #1: Yes

Reviewer #2: Yes

4. Have the authors made all data underlying the findings in their manuscript fully available?

Reviewer #1: Yes

Reviewer #2: Yes

5. Is the manuscript presented in an intelligible fashion and written in standard English?

Reviewer #1: Yes

Reviewer #2: Yes

6. Review Comments to the Author

Reviewer #1: Overall, the authors appropriately addressed my comments; and the quality of the manuscripts has greatly improved. There still exist some points need to be fixed:

1) It seems that the ground truth (HR) projections in training the VDSR network is the original CDT acquisitions. If this is the case, the importance of the whole work would be dramatically decreased. Since the LR projections are all artificially down sampled, the clinical importance of the proposed workflow would land on obtaining low resolution projections to reduce imaging noise and time. In addition, all the result images should be compared with the reconstruction from the original projections with just the VDSR being excluded. The authors should add comments on this.

2) The total acquisition time is 6.4s. Then if we want to clinically apply this workflow, the patients should hold their breath for such a time. Some comments about the challenges about the anticipated clinical application should also be added.

3) According to Fig. 8(c), the combinations of 256/10 and 512/40 have the highest SDNR, why did you choose 128/70?

4) The Introduction seems still lengthy and need to be logically improved. Please try to re-read it and make some revision.

5) Some grammar issues still can be found, such as lines 464-466: “NPS acquired uniform water phantom … and analyzed them by using the two-dimensional Fourier analysis method [54] using an in-focus plane.”

Reviewer #2: (No Response)

7. PLOS authors have the option to publish the peer review history of their article (what does this mean?). If published, this will include your full peer review and any attached files.

Reviewer #1: No

Reviewer #2: No

---

## [Author Response · Author response to Decision Letter 1]

12 Dec 2020

Thank you in advance for your kind consideration of this paper. I attach here our revised manuscript, as well as a point-by-point response to the reviewer’s comments.

We feel that the 2th revised manuscript is a suitable response to the comments, and is significantly improved over the 1th revised submission. 

Sincerely yours,

Tsutomu Gomi, Kitasato University

---

## [Editor Report · Decision Letter 2]

16 Dec 2020

Improved digital chest tomosynthesis image quality by use of a projection-based dual-energy virtual monochromatic convolutional neural network with super resolution

PONE-D-20-19441R2

Dear Dr. Gomi,

We’re pleased to inform you that your manuscript has been judged scientifically suitable for publication and will be formally accepted for publication once it meets all outstanding technical requirements.

Kind regards,

Jeonghwan Gwak, PhD

Academic Editor

PLOS ONE

Additional Editor Comments (optional):

All of the reviewers' concerns have been addressed, and it is now suitable for publication.
---

## [Editor Report · Acceptance letter]

21 Dec 2020

PONE-D-20-19441R2 

Improved digital chest tomosynthesis image quality by use of a projection-based dual-energy virtual monochromatic convolutional neural network with super resolution 

Dear Dr. Gomi:

I'm pleased to inform you that your manuscript has been deemed suitable for publication in PLOS ONE. Congratulations! Your manuscript is now with our production department. 

Kind regards, 

on behalf of

Dr. Jeonghwan Gwak 

Academic Editor

PLOS ONE